# The Benefit of Multigene Panel Testing for the Diagnosis and Management of the Genetic Epilepsies

**DOI:** 10.3390/genes13050872

**Published:** 2022-05-13

**Authors:** Heather Leduc-Pessah, Alexandre White-Brown, Taila Hartley, Daniela Pohl, David A. Dyment

**Affiliations:** 1Division of Neurology, Department of Pediatrics, Children’s Hospital of Eastern Ontario, University of Ottawa, Ottawa, ON K1H 8L1, Canada; dpohl@cheo.on.ca; 2Department of Genetics, Children’s Hospital of Eastern Ontario Research Institute, University of Ottawa, Ottawa, ON K1H 8L1, Canada; awhitebrown@cheo.on.ca (A.W.-B.); thartley@cheo.on.ca (T.H.); ddyment@cheo.on.ca (D.A.D.)

**Keywords:** epilepsy panel, genetic epilepsy, epilepsy, epileptic encephalopathies

## Abstract

With the increasing use of genetic testing in pediatric epilepsy, it is important to describe the diagnostic outcomes as they relate to clinical care. The goal of this study was to assess the diagnostic yield and impact on patient care of genetic epilepsy panel testing. We conducted a retrospective chart review of patients at the Children’s Hospital of Eastern Ontario (CHEO) who had genetic testing between the years of 2013–2020. We identified 227 patients that met criteria for inclusion. The majority of patients had their testing performed as “out-of-province” tests since province-based testing during this period was limited. The diagnostic yield for multi-gene epilepsy panel testing was 17% (39/227) and consistent with the literature. Variants of unknown significance (VUS) were reported in a significant number of undiagnosed individuals (77%; 128/163). A higher diagnostic rate was observed in patients with a younger age of onset of seizures (before one year of age; 32%; 29/90). A genetic diagnosis informed prognosis, recurrence risk counselling and expedited access to resources in all those with a diagnosis. A direct change in clinical management as a result of the molecular diagnosis was evident for 9% (20/227) of patients. The information gathered in this study provides evidence of the clinical benefits of genetic testing in epilepsy and serves as a benchmark for comparison with the current provincial Ontario Epilepsy Genetic Testing Program (OEGTP) that began in 2020.

## 1. Introduction

In the past decade, the use of next generation sequencing has transformed the field of epilepsy genetics. There has been an exponential increase in the rate of discovery of epilepsy-associated genes [1,2]. These genes can now be routinely sent for sequencing from out-patient neurogenetic or epilepsy clinics. Genetic results provide improved care with informed prognoses, appropriate surveillance measures, and treatment optimization [3,4]. In addition, genetic diagnoses limit further unnecessary investigations and provide psychosocial benefits to patients and their families [5,6].

Clinical genetic testing for epilepsies can include microarray, single or targeted gene testing, multi-gene panels, or whole exome sequencing. Multi-gene epilepsy panels are often a first-line approach to interrogate genes known to be associated with epilepsy. However, there are a wide variety of multi-gene panel tests to choose from. Prior to 2020, a multi-gene panel was not offered by a clinical laboratory within the province of Ontario. Testing was necessarily pursued “out-of-province”, typically in commercial laboratories located in the United States [7]. Ontario and the other provinces in Canada have a single payer health care system and testing in Ontario is funded by the provincial Ministry of Health (MOH). The growing availability of epilepsy panel options resulted in increased costs and inconsistencies in access to testing across the province [7]. In 2014, the Ministry of Health of Ontario established the Genetic Testing Advisory Committee (GTAC) to review genetic testing and establish guidelines for the publicly-funded use of panel testing in epilepsy [7]. This report provided guidance for patient selection and appropriate testing modalities [7]. A provincial Ontario Epilepsy Genetic Testing Program (OEGTP) was implemented making testing available within the province since October 2020 [8].

We sought to characterize the results of multi-gene epilepsy panel testing for patients with epilepsy at a regional tertiary care hospital, The Children’s Hospital of Eastern Ontario (CHEO), prior to the implementation of within province testing. Specifically, we report the diagnostic rates for epilepsy panel testing and the impact of the molecular diagnoses on patient management, in addition to identifying the patient demographics in which a molecular diagnosis is more likely to be identified. This serves as a benchmark for future comparison of the outcomes associated with genetic testing for epilepsy in Ontario.

## 2. Methods

### 2.1. Inclusion Criteria and Cohort Identification

This was a retrospective chart review of patients seen at CHEO between January 2013 and December 2020. CHEO’s catchment areas includes Eastern Ontario, Western Quebec, parts of Northern Ontario, and Nunavut. Subjects were identified based on diagnostic codes in our electronic medical records (EPIC Systems) including all diagnoses for seizures and epilepsy (ICD-10-CA: G40.** and ICD-10-CA: R56.**) AND the presence of any genetic test order. The identified charts were then manually reviewed to confirm eligibility for this study based on the presence of a completed multi-gene epilepsy panel test. Charts of patients with seizures or epilepsy with whole exome sequencing and without panel testing were also identified from the search of our medical records and reviewed. Local research ethics board approval was obtained for this study (CHEO REB protocol No. 21/33X).

### 2.2. Chart Review

Basic demographic information including partial date of birth, age at testing, and test ordered were automatically pulled from the electronic medical record. Manual chart review was performed to confirm eligibility for the study and to extract additional demographic information. All information was coded and medical record numbers were not included in the data collection tool. Study data were collected and managed using REDCap electronic data capture tools hosted at CHEO [9,10]. REDCap (Research Electronic Data Capture) is a secure web-based software platform designed to support data capture for research. Information collected included basic demographics, epilepsy history, co-morbidities, genetic testing results and follow-up care. The data were collected at one static time point (time of chart review). Data were reported as number of patients with specific characteristics or percentage of the total population.

### 2.3. Data Analysis

Data were reported as number of patients with specific characteristics or percentage of the total population. Data were collected and analyzed using Graphpad Prism version 6.07 software. Diagnostic yield was calculated for various subgroups of patients and compared to overall diagnostic rate. Data were analyzed using standard statistical tests (Student *t*-test, chi-squared). Significance was adjusted for multiple comparisons and considered statistically significant for *p*-values < 0.05.

## 3. Results

The search of the electronic medical records identified 699 patients with a diagnosis of epilepsy or seizures, with any genetic testing. Of these, 227 had multi-gene epilepsy panel testing, 96 patients had whole exome sequencing without multi-gene epilepsy panel testing and the remaining 376 had microarray only, targeted gene testing, or other panel tests (for example, Intellectual disability panel) (Figure 1). We used the multi-gene epilepsy panel cohort of 227 patients for the analysis reported in this study. The cohort included approximately half males (117; 51%) and females (110; 49%) (Table 1). The ethnic composition of the retrospective composition was predominantly of European ancestry (197 (87%) European; 10 (4%) Middle Eastern; 7 (3%) African; 7 (3%) South-East Asian; 3 (1.3%) Asian and 3 (1.3%) Indigenous). The multi-gene epilepsy panels were performed in different laboratories outside of Ontario (Table 1) and consisted of a varying content of epilepsy-associated genes (ranging from 10 genes to 902 genes).

In addition to the multi-gene panel testing, 66% (149/227) of patients had a microarray performed and 21% (47/227) had subsequent whole exome sequencing (Table 1). As part of their medical work-up, 91% (207/227) of patients had an MRI and 100% (227/227) had an EEG (Table 1). Co-morbidities, family history and response to anti-seizure medications are reported in Table 1. Epilepsy panel testing revealed a genetic diagnosis in 17% (39/227) of all patients tested (Figure 2A). A genetic diagnosis was considered positive if the variant was reported as pathogenic, or if reported as likely pathogenic or variant of uncertain significance (VUS) and evaluation by a geneticist confirmed the mutation was suspected to be causing the associated symptoms (for example, if parental testing was subsequently performed and the variant was found to be de novo). Genetic diagnoses and specific variants are reported in Figure 2B and Table A1. The most frequent gene with pathogenic or likely pathogenic variation was *SCN1A* (6 patients), followed by *KCNQ2*, *PCDH19* and *PRRT2* (4 patients each). A variety of additional pathogenic variants were identified in genes for which 1 or 2 patients were positive (Figure 2B). All VUSs included as “positive” results were reviewed for this study and rationale for pathogenicity is reported in Table A2 and consistent with guidelines [11]. Subsequent genetic testing in the remaining 188 individuals diagnosed 10% of patients via microarray (*n* = 2), whole exome sequencing (*n* = 7), or other targeted gene or panel testing (*n*= 13) (Figure 2A,C,D). Variants of uncertain significance (VUS) were reported in 77% of patients without a genetic diagnosis (Figure 2E).

A direct change in medical management as a result of testing was recommended for 9% (20/227) of patients who were tested by multi-gene epilepsy panel (Figure 3A). This correlates to over 50% (20/39) of the patients with a genetic diagnosis (Figure 3A). Direct changes to medical management included anti-seizure medication adjustments and surveillance via investigations or referrals for disease-associated co-morbidities (Figure 3B).

We next analyzed sub-groups of patients to determine diagnostic rates of panel testing. The diagnostic rate of epilepsy panel testing was significantly higher (32%; 29/90) in patients with onset of epilepsy under 1 year of age (Figure 4A). The diagnostic rate declines with seizure onset >1 year of age with a rate of 12% (9/78) from ages 1–3 and a rate of 1% (1/85) when seizures began >3 years of age (Figure 4A). Epileptic encephalopathies (32%) and autism spectrum disorder (28%) were the co-morbidities associated with the highest diagnostic rates although the rate was only significant higher in the epileptic encephalopathy group (Figure 4B). With a family history of epilepsy or seizures, the diagnostic rate was 22% (16/74) (Figure 4B).

Of the 699 patients originally identified in our study, 96 had WES without a multi-gene epilepsy panel (Figure 1). In order to qualify for WES individuals had to meet at least two of the following clinical criteria for publicly-funded testing: (1) have multisystem involvement, (2) moderate to severe developmental or functional impairment, (3) progressive clinical course, (4) the differential diagnosis includes two or more well defined conditions requiring evaluation by multiple targeted gene panels, and (5) suspected severe genetic syndrome NYD for which multiple family members are also affected or when parents are consanguineous. Of the 96 participants that had exome sequencing, developmental delay and/or intellectual disability were common (80%; 77/96) (Table 2). The diagnostic rate of WES in patients with epilepsy meeting the above criteria was 40% (38/96; Figure 4C). In an additional 13% (13/96) of these patients, a compelling variant in a novel disease gene of uncertain clinical significance (GUS) to explain their clinical presentation was identified but could not be confirmed at the time of the report (Figure 4C). The mean age of seizure onset was higher (4.2 years) in the WES cohort compared to the panel cohort (2.5 years) (Table 1 and Table 2). Despite a mean age of seizure onset of 2.5 years, panel testing was carried out at a mean age of 6.5 years (Figure 5B, Table 1). Similarly, in the WES cohort, testing was carried out later at a mean of 8.7 years where the mean age of seizure onset was 4.2 years (Table 2). The average interval between age of seizure onset and age of testing “delay to testing” was 4.0 years in the panel cohort. Overall, the delay to testing was shorter in more recent years. Between the years of 2013–2016 the delay to testing was 5.2 years and between 2017–2020 it was 3.1 years.

The OEGTP pre-requisite check-list requires MRI and EEG prior to multi-gene panel testing (Available at: https://www.lhsc.on.ca/palm/docs/OEGTP%20Requisition.pdf) (accessed on 8 April 2022). While all patients in our cohort had an EEG, 9% (20/227) did not have an MRI prior to genetic testing (Table 1, Figure 5C). The diagnostic rate for patients with MRI was 19% and for those without MRI prior to testing was 5% (Figure 5C). Those without an MRI had generalized seizures including myoclonic, absence and generalized tonic-clonic seizures. We compared the annual diagnostic rate of panel testing over the course of our study and observed an increase in diagnostic yield between 2014–2016, and a stable diagnostic rate between 2016–2020 (Figure 5D). We compared the diagnostic rate based on panel size and saw highest rates (20%) in panels containing 100–299 genes (Figure 5E).

## 4. Discussion

### 4.1. Panel Diagnostic Rate

In this study we provide a representative overview of the yield of epilepsy panel testing at our institution over an eight year period. Our diagnostic yield of 17% is comparable to other reported epilepsy panel diagnostic rates [12,13,14] including a recent meta-analysis of 31,000 patients which reported a diagnostic yield of 19% for multi-gene panel testing [2]. In this comprehensive meta-analysis, Sheidley et al., also reported that increased panel size correlated with increased diagnostic yield [2]. The panels included in our cohort ranged from 10–902 genes, with an average of 200 genes. Smaller panels include those specific to an epilepsy subgroup such as neonatal onset, stat panel for treatable conditions, or progressive myoclonic epilepsy. With the inclusion of all panels, our results may represent an underestimation of the genetic epilepsies as some diagnoses may have been missed despite no clear correlation of panel size with diagnostic yield. Diagnostic yield was highest in panels containing 100–299 genes; however, with panel size ≥300, diagnostic rate was lower at 14%. In addition, further diagnoses will be made as the field evolves and more information becomes available to better interpret variants, and identify new epilepsy-associated genes. Patients highly suspected to have an underlying genetic diagnosis or with identified variants are often re-analyzed every 1–3 years to determine if there is any change in the status of their results. Variants of uncertain significance were noted in 77% of undiagnosed patients in our cohort, some of which will be re-classified as pathogenic over time.

### 4.2. Changes to Patient Management as a Result of Testing

A genetic diagnosis leads to changes in patient management in an increasing number of diagnoses [15,16]. We found that all patients benefited from a genetic diagnosis and that there was a direct impact on medical management in 9% of those tested. Our results are comparable to other recent studies reporting changes in management of 5–13% [17,18]. A direct impact on medical management included recommended anti-seizure medications and additional investigations, referrals or follow-up for associated conditions. The largest group of actionable diagnoses were for patients with *SCN1A* mutation for whom sodium channel blocking agents are contraindicated [16,19,20]. On the contrary, sodium channel blocking agents are recommended for patients with *SCN8A*, *KCNQ2*, *KCNQ3* mutation-associated epilepsies [16,21] p. 8, [22,23] p. 3. Those with *SCN8A* mutations are also screened for hearing and visual deficits and monitored for associated movement disorders [24,25], p. 8. With *PRRT2*-associated epilepsies, both a medication change and additional surveillance are recommended. These patients are noted to respond well to carbamazepine and are monitored and counselled for the risk of developing paroxysmal movement disorders and hemiplegic migraines [26]. Patients with tuberous sclerosis are routinely monitored for multi-system involvement and risk of malignant transformation [27]. Surveillance for patients with *PACS1* mutations includes monitoring for growth and failure to thrive, and screening for structural malformations of the eyes, brain, heart and kidneys [28], p. 1. Finally, *SLC6A1* mutations are associated with co-morbid autism spectrum disorder (ASD), learning disabilities and ADHD [29]. While valproic acid has been reported to be effective in some cases of *SLC6A1* associated epilepsy, there is minimal evidence to suggest it’s efficacy over other anti-seizure medications [30], p. 6. Recommendations for other genetic epilepsies from case reports or anecdotal evidence have not been included in our quantification of changes to medical management; however, they may still help guide medical management on a case by case basis. A diagnosis also provided the ability to prognosticate patient outcomes. This includes early differentiation between self-limited epilepsies of infancy and syndromes known for refractory epilepsy and severe developmental delays. This distinction can be clinically impossible at early assessments. After a genetic diagnosis, many families are offered parental testing in order to determine risk of recurrence for other children. In our study, 64% of families with a positive diagnosis pursued parental testing (Table 1). Families were also advised on available resources (i.e., organizations specific to the diagnosis), and were able to seek out family support groups. Additionally, with a confirmed diagnosis, access to funding for supportive resources was facilitated for therapies such as occupational and physical therapy, as well as respite.

### 4.3. Delay to Panel Testing

The majority of epilepsies begin in childhood with onset at less than one year of age commonly observed, particularly for genetic epilepsies [31,32,33]. Seizure onset occurred before one year of age for 40% of the patients in our multi-gene panel cohort (Figure 5A, Table 1). However, the mean age at panel testing was much later at 6.5 years of age (Figure 5B, Table 1). This delay between onset of seizures and timing of panel testing is likely multi-factorial in nature. The lack of available panel testing at the time of initial assessment would have led to delays for the older patients in our cohort. We did notice an overall average shorter interval between age of seizure onset and age of testing for panels run between 2017–2020 (3.1 years) compared to 2013–2016 (5.2 years). The progression of the patient’s clinical course (which was not reviewed in detail in our study) would have also impacted the indication for genetic testing. Lastly, the need to apply to the Ministry of Health for approval of out-of-province funding could have delayed testing. Going forward, these delays would be expected to be shortened given the availability of local testing and expert knowledge of the variation in the associated epilepsy genes. Nevertheless, the delays in testing observed will serve as a useful benchmark for the success of future Ontario-based testing.

### 4.4. Limitations

The inability to control for patient selection remains a significant limitation to our study. Since not all patients with epilepsy were sent for genetic testing, decisions were based on the clinical judgement and experience of the ordering physician. Prior to the guidelines for genetic epilepsy testing in Ontario published online in 2016 [34], there was minimal formal guidance [7,35]. With a lack of a standardized approach, physicians used their own practices for testing. Anecdotally, the decision to send for testing was often based on age of onset of seizures, co-morbidities, family history or clinical presentation. The total 227 comprised varied panels with differences in the number of genes as well as the methods used (for example, some tests could capture exon-level del/dup and others could not). This degree of variability with the panel test performed needs to be considered when interpreting these combined results. Sample size was limited for comparisons within the cohort. When multiple comparisons were considered, only diagnoses within the DEE group (compared to those without DEE) remained significant.

It is important to note that the results described in this study capture a representative sample of epilepsy panel testing at a single tertiary care hospital in Ontario. However, a number of genetic diagnoses are made at our centre by targeted genetic testing (such as for *GLUT1*, *SCN1A* or *TSC*) that are not captured in this cohort of 227 patients who had multi-gene panel testing. In addition, while the electronic search of our medical records facilitated the collection of a large number of charts over 8 years it relies on the correct diagnostic codes and orders to be entered, likely leading to occasional missed records.

### 4.5. Implications

As more genetic epilepsies are characterized, the yield of genetic testing will continue to rise along with the impact on medical management of patients. With the implementation of guidelines for publicly funded intra-provincial epilepsy panel testing in Ontario there may be an increase in the number of patients sent for genetic testing. With this guidance, pediatricians and general practitioners may be better able to identify individuals eligible for genetic testing. This will also reduce the delay between panel testing and seizure onset by eliminating the need to wait for a specialist referral. Recent initiatives such as ECHO: Epilepsy across the lifespan would also facilitate informed testing by care providers across the province [36]. Furthermore, with the use of a standardized approach to genetic panel testing in epilepsy based on expert consensus and the identification of new epilepsy-associated genes, we predict diagnostic yield will also increase. The outcomes reported in this study will serve as a benchmark and allow for ongoing evaluation of genetic panel testing for epilepsy to assess the impact of these changes. Furthermore, the results reported in our study can inform families of the current expectations for testing outcomes as they relate to diagnosis and medical management changes.

## 5. Conclusions

Panel testing had a diagnostic yield of 17% and led to changes in medical management for 9% of all patients tested. Changes to medical management included targeted treatment recommendations and surveillance for co-morbidities. Prognostic information and genetic counselling for recurrence risk were provided to all patients that received a diagnosis. A younger age of onset of seizures and epileptic encephalopathy are associated with higher diagnostic rates. With the implementation of guidelines for Ontario-based testing following our study we predict a further increase in the diagnostic yield of epilepsy testing. Our results will help guide the selection of patients for testing and the ongoing evaluation of testing criteria. Furthermore, advances in genetic testing and characterization of genetic epilepsies will continue to increase the impact of multi-gene epilepsy panel testing.

## Figures and Tables

**Figure 1 genes-13-00872-f001:**
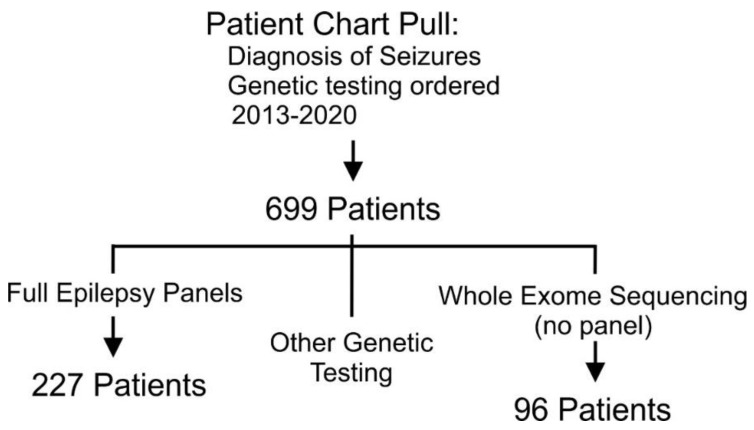
Methods. Parameters of medical record identification from our electronic medical record system (EPIC Systems). Charts were selected based on ICD diagnostic codes including ‘seizures’ or ‘epilepsy’ and any order under ‘genetic testing’. We queried our electronic records from 2013 to the end of 2020. 699 patients were identified in the original chart pull, 227 of whom met our inclusion criteria with a multi-gene epilepsy panel sent and resulted and 96 of whom had whole exome sequencing without an epilepsy panel.

**Figure 2 genes-13-00872-f002:**
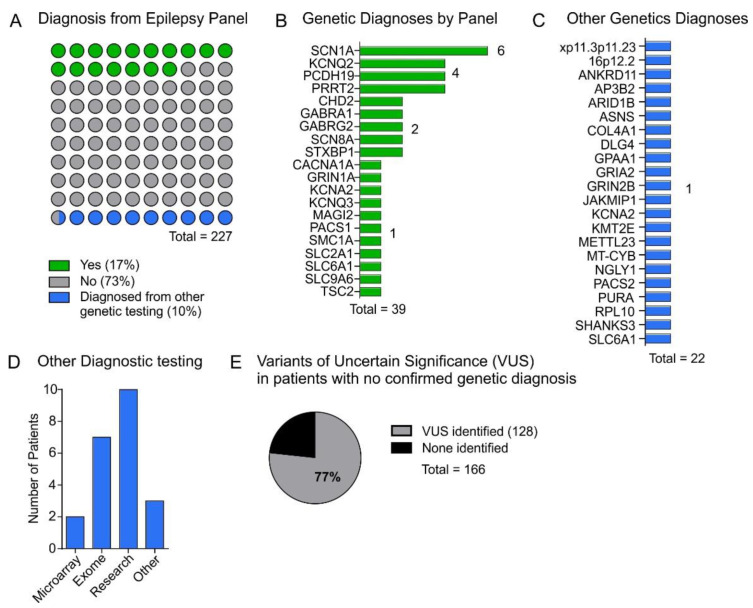
Panel testing results. (**A**) Dot plot representing overall epilepsy panel testing results (*n* = 227). The panel test identified a genetic diagnosis “yes” in 17% of patients tested and 10% were diagnosed from other genetic testing carried out after the panel test. (**B**) Epilepsy-causing genetic mutations identified by panel testing. (**C**) List of genetic mutations identified by other genetic testing. (**D**) Other genetic tests that resulted in a diagnosis which explained the patient’s epilepsy phenotype. Research testing included research exomes and further targeted analysis, other testing including mitochondrial testing and other non-epilepsy panels (**E**) Of the 166 (73%) patients in our cohort without an identified genetic diagnosis 77% had variants of uncertain significance (VUS) identified.

**Figure 3 genes-13-00872-f003:**
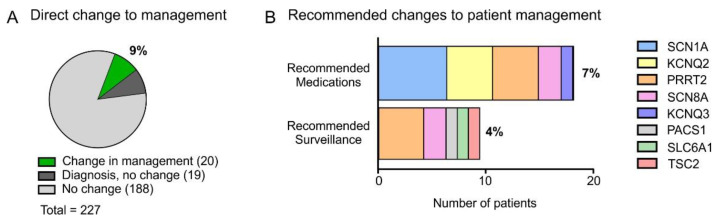
Changes in management. (**A**) Percentage of all patients tested for whom genetic diagnosis resulted in a change in medical management (9%). Green and grey sections combined represent the patients that received a genetic diagnosis from panel testing (17%) with the green portion showing that approximately half of those diagnosed had changes in management. (**B**) The genetic mutations associated with recommendations for changes in medical management including anti-seizure medication recommendations and surveillance or screening for co-morbidities.

**Figure 4 genes-13-00872-f004:**
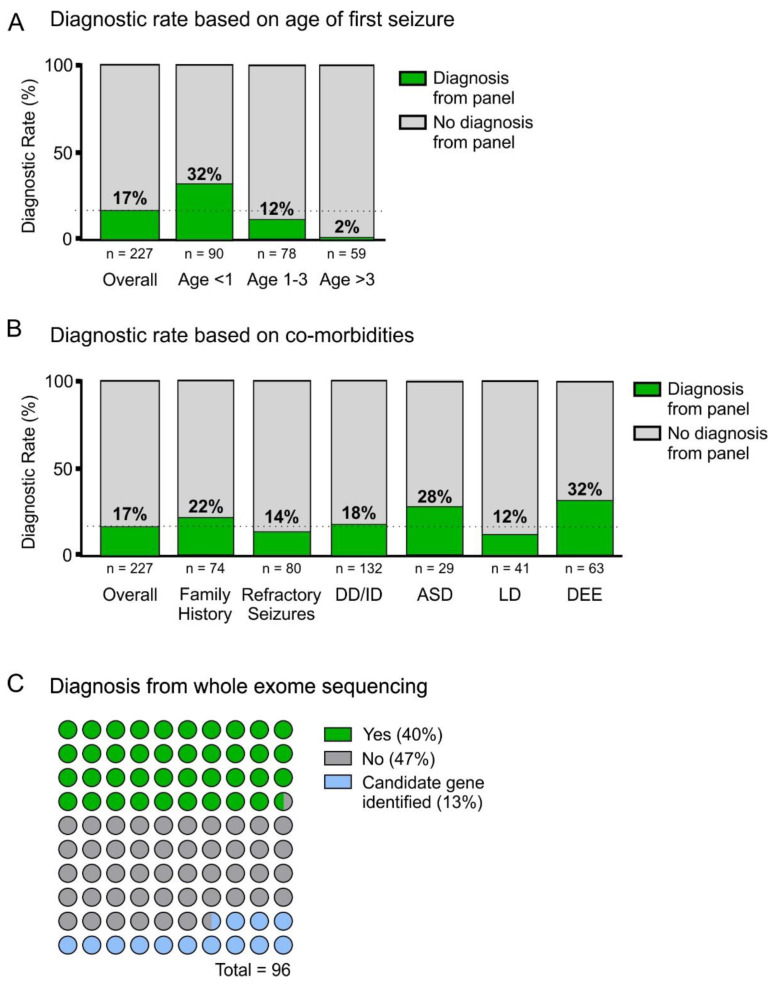
Diagnostic rate in patient sub-groups. (**A**,**B**) Diagnostic rate of epilepsy panel testing comparisons with overall rate of 17% (dotted line) grouped by age of onset of seizures. (**A**) All age groupings; comparison among groupings showed a significant difference, χ^2^ (2, *n* = 227) = 25.99, *p* < 0.0001. (**B**) Frequency of diagnoses by co-morbidities. For family history, χ^2^ (1, *n* = 227) = 5.57, *p* = 0.018; Refractory seizures χ^2^(1, *n* = 227) = 0.41, *p* = 0.52; DD/ID χ^2^ (1, *n* = 227) = 0.686, *p* = 0.41; ASD χ^2^ (1, *n* = 227) = 2.53, *p* = 0.112; LD χ^2^ (1, *n* = 227) = 0.874, *p* = 0.35; DEE χ^2^ (1, *n* = 227) = 13.00, *p* = 0.0003. Only DEE was significant after consideration of multiple comparisons. (**C**) Dot plot representing results of whole exome sequencing (WES, *n* = 96). WES identified a genetic diagnosis in 40% of patients tested and a candidate VUS in an additional 13%.

**Figure 5 genes-13-00872-f005:**
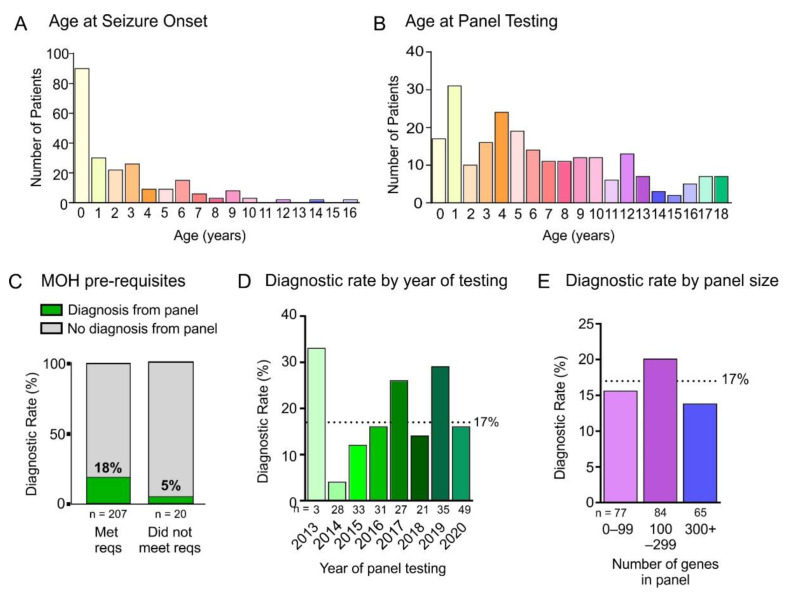
Guidelines. (**A**,**B**) Age of patients at seizure onset (**A**) and age at panel testing (**B**) for the patients in the epilepsy panel cohort. (**C**) Diagnostic rate of patients who met the Ministry of Health of Ontario testing pre-requisite investigations of EEG and MRI compared to those who did not meet the requirement (no MRI head was carried out). (**D**) Diagnostic rate of epilepsy panel testing by year of testing compared to overall rate of 17% (dotted line). (**E**) Diagnostic rate of epilepsy panel testing based on number of genes in epilepsy panel.

**Table 1 genes-13-00872-t001:** Summary of results of epilepsy panel cohort.

	All Panel Testing *n* (%)	Positive Diagnosis *n* (%)
Participants	227	39
Sex (%)		
Male	117 (51)	19 (49)
Female	110 (49)	20 (51)
Panel Company		
GeneDx	92 (40.5)	17 (43)
Fulgent	45 (20)	7 (18)
MNG	20 (9)	5 (13)
Invitae	27 (12)	4 (10)
Blueprint Genetics	14 (6)	3 (8)
Prevention Genetics	8 (3.5)	1 (2.5)
Transgenomics	13 (6)	0 (0)
Other	8 (3)	2 (5)
Age at seizure onset (years)		
0	90 (40)	29 (74)
1	30 (13)	4 (10)
2	22 (10)	5 (13)
3	26 (11)	0 (0)
4	9 (4)	0 (0)
≥5	50 (22)	1 (2.5)
Mean	2.5	0.5
Age at testing (years)		
0	17 (7)	8 (20.5)
1	31 (14)	10 (26)
2	10 (4)	1 (2.5)
3	16 (7)	4 (10)
4	24 (11)	3 (8)
≥5	129 (57)	13 (33)
Mean	6.5	4.5
Investigations		
Microarray	149 (66)	23 (59)
Parental testing	92 (41)	25 (64)
Whole exome sequencing	47 (21)	0 (0)
MRI	207 (91)	38 (97)
EEG	227 (100)	39 (100)
Co-morbidities and family history		
Refractory Seizures	80 (35)	11 (28)
Developmental Delay/Intellectual Disability	132 (58)	24 (62)
Autism Spectrum Disorder	29 (13)	8 (20.5)
Learning Disability	41 (18)	5 (13)
Dysmorphisms or multi-system involvement	33 (15)	3 (8)
Epileptic encephalopathy	63 (28)	19 (49)
Family History of Epilepsy	74 (33)	16 (41)

**Table 2 genes-13-00872-t002:** Summary of results of whole exome sequencing cohort.

	All Exome Testing *n* (%)	Positive Diagnosis *n* (%)
Participants	96	38
Age at seizure onset (years)		
0	18 (19)	8 (21)
1	9 (9)	6 (16)
2	12 (13)	3 (80)
3	8 (8)	3 (8)
4	12 (13)	6 (16)
≥5	27 (28)	9 (24)
Unknown	9 (9)	3 (8)
Mean	4.2	4.1
Age at testing (years)		
0	4 (4)	2 (5)
1	7 (7)	4 (11)
2	4 (4)	2 (5)
3	8 (8)	3 (8)
4	8 (8)	3 (8)
5–8	13 (14)	4 (11)
≥9	52 (54)	20 (53)
Mean	8.7	8.6
Co-morbidities		
Developmental Delay/Intellectual Disability	77 (80)	34 (89)
Autism Spectrum Disorder	18 (19)	6 (15)
Learning Disability	2 (2)	0 (0)
Dysmorphisms or multi-system involvement	96 (100)	38 (100)

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
