# Peer review of "The Benefit of Multigene Panel Testing for the Diagnosis and Management of the Genetic Epilepsies"

_genes, 2022, doi:10.3390/genes13050872_

Round 1
Reviewer 1 Report
The study by Leduc-Pessah et al. reports the benefits of multigene testing for genetic epilepsies and may help in gathering readers attention. However, I have some concerns about the methods description and analysis performed by the authors.
- The authors need to provide more details on their methodology. Since they are doing a genetic study, it is important to report and analyse the ethnicity of the subjects. They also need to mention the multigene tests as there are a variety of them available. Were they different? Did they all include same number of genetic variants?
- The authors are suggesting that association of early age at onset with the diagnostic rate. However, I don't see any statistical analysis done to confirm this. It would be better if the level of significance (p-value) is reported in the results section.
- The sample size also seems small. Can the authors provide the power of the study? If it is underpowered, it should be reported as a study limitation.
Overall, the study is well-written, however, requires a major revision to increase its impact.
Author Response
Response to Reviewer 1
We thank the reviewer for their comments. The suggested revisions have helped to increase the clarity and impact of our study.
1. The authors need to provide more details on their methodology. Since they are doing a genetic study, it is important to report and analyse the ethnicity of the subjects. They also need to mention the multigene tests as there are a variety of them available. Were they different? Did they all include same number of genetic variants?
We have expanded the details in the methods section of the manuscript (Page 2). All patients seen at CHEO between 2013-2020 meeting the inclusion criteria were included in the study. While patients seen at CHEO span a diverse population from CHEO’s catchment areas including Eastern Ontario, Western Quebec, parts of Northern Ontario and Nunavut. In general, the majority state they are of European descent (>85%). We have included a statement on the catchment area of CHEO in the methods (Page 2, Line 62) and a comment on ethnicity in the Results.
In order to increase our sample size we have included patients with all multi-gene epilepsy panels. Multigene panels include comprehensive epilepsy panels, infantile epilepsy panels, myoclonic epilepsy panels, and epileptic encephalopathy panels (Page 11, lines 228-231). Number of genes ranged from 10-902 (Page 11, line 229). Panels were sent to multiple different companies for processing (Table 1). We have discussed the limitations of including this range of panels in the discussion (Page 11-12, lines 231-235). The inconsistencies observed among the ordering providers was one of the many reasons for the province of Ontario to repatriate this testing.
2. The authors are suggesting that association of early age at onset with the diagnostic rate. However, I don't see any statistical analysis done to confirm this. It would be better if the level of significance (p-value) is reported in the results section.
We have included the statistical testing in the results section p<0.0001 (Figure 4A, Page 8, Line 173).
3. The sample size also seems small. Can the authors provide the power of the study? If it is underpowered, it should be reported as a study limitation.
As this was a retrospective, descriptive chart review without a defined control sample for comparison, the power analysis was limited. Nevertheless, as per the Reviewers comment, we have now commented on this within the limitations of the study (Page 13, Lines 302-307).
Reviewer 2 Report
Leduc-Pessah et al. present a well-written manuscript including the results of epilepsy gene panel testing in a cohort of 227 patients with epilepsy. This is a retrospective study including multiple different panel designs in patients where a clinical decision was made to perform genetic testing. The study showed a diagnostic yield of 19% with effects on clinical management in 9%. These findings align with prior studies on this topic. The study also provides information on a cohort of 96 patients who directly underwent exome sequencing with a diagnostic yield of 40%. The findings are principally of interest given the differences in the study cohort and gene panels used in the different studies. However, there are some limitations to the study which primarily stem from the retrospective nature and use of different gene panels throughout the duration of the study. Furthermore, the decision to go ahead with genetic testing was made by the responsible clinician without clear inclusion criteria which makes the results difficult to apply to different cohorts.
Other items:
- The authors state that they also considered VUS as a positive result if a geneticist confirmed that the variant was suspected to be causing the patient's symptoms. I am concerned that this may have resulted in the inclusion of uncertain variants into the diagnostic yield. With proper workup including segregation testing it should be possible to upgrade VUS to likely pathogenic. For example, the CACNA1A variant was described as de novo. Did this really remain a VUS according to ACMG criteria? If variants cannot be classified as likely pathogenic, they should be considered as VUS and not reported as positive. The authors could report these as suspicious VUS but they need to be reported separately.
- Could the authors specify if the number of genes included in the different panels correlated with diagnostic yield?
- I am wondering if the different yield over the years was related to number of genes on the panel as typically gene content of panels increased over time. This should be discussed and data provided.
- The authors describe a significant delay between seizure onset and the time of panel testing. As discussed by the authors this could well relate to the fact that panel testing was not available before. Did this delay decrease over the time of the study as presumably more patients with new onset seizures were included?
- The authors should either also provide the genomic position of the variants or the transcript accession number that was used for describing the variant.
Round 2
Reviewer 1 Report
The authors have improved the manuscript as per the changes suggested earlier.
Reviewer 2 Report
The authors have successfully addressed the issues raised by the reviewers. I have no further comments.